



# Weight-to-weight conversion factors for benthic macrofauna: recent measurements from the Baltic and the North Seas

Mayya Gogina[1]★, Anja Zettler[1]★, Michael L. Zettler[1]

[1]Biological Oceanography, Leibniz Institute for Baltic Sea Research Warnemünde, Seestraße 15, 18119 Rostock, Germany

*Correspondence to*: Mayya Gogina (mayya.gogina@io-warnemuende.de)

★ These authors contributed equally to this work.

**Abstract.** Estimates of biomass often involve the use of weight-to-weight conversion factors for rapid assessment of dry-weights based metrics from more widely available measurements of wet weights. Availability of standardized biomass data is essential amid research on population dynamics, energy flow, fishery and food web interactions. However, for many
species and groups the widely-applicable freely available conversion factors until now remained very rough approximations with high degree of taxonomic generalization. To close up this gap, here for the first time we publish the most detailed and statically robust list of ratios of wet weight (WW), dry weight (DW) and ash-free dry weight (AFDW). The dataset includes over 17000 records of single measurements for 497 taxa. Along with aggregated calculations, enclosed reference information with sampling dates and geographical coordinates provides the broad opportunity for reuse and repurposing. It empowers the
future user to do targeted sub selections of data to best combine them with own local data, instead of only having a single value of conversion factor per region. Data can help to quantify natural variability and uncertainty, and assist to refine current ecological theory. The dataset is available via an unrestricted repository from: http://doi.io-warnemuende.de/10.12754/data-2021-0002 (Gogina et al., 2021).

## 1 Introduction

Research on energy flow, food web interactions, fishery and population dynamics, and role of biodiversity in ecosystem functioning depend on the estimates of biomass and secondary production. This broad range of studies often involve the use of weight-to-weight conversion factors for rapid assessment of required dry-weights based metrics from less time-consuming and therefore more widely available wet weights biomass measurements (e.g. Ricciardi and Bourget, 1998 and references therein; Gogina et al., 2020). Conversion factors derived from subsamples to enable data standardisation and determination
of dry weight for large volume of material. With growing interest in biodiversity in the second half of the last century, primarily efforts from the Baltic Sea pioneered publishing the compilations of conversion factors for marine macroinvertebrates (Thorson, 1957; Lappalainen and Kangas, 1975; Rumohr et al., 1987), that later expanded to other geographic regions (Petersen and Curtis, 1980; Tumbiolo and Downing, 1994; Ricciardi and Bourget, 1998; Brey et al.,

2010). However, though allow general biomass estimates, for many species and groups the available widely-applicable conversion factors for data standardization remain very rough approximations of weight-to-weight relationships. For example, the global database for meio-, macro- and megabenthic biomass and densities that was recently published by Stratmann et al. (2020) includes only little share of measured ash-free dry weights and cites only a handful of publications (including those listed above) that provide such broadly used sets of values for the corresponding conversion. This highlights the importance of presented compilation.

Here for the first time we publish the taxonomically most detailed and statically most robust list of ratios of wet weight (WW), dry weight (DW) and ash-free dry weight (AFDW) based on over 17.000 measurements for 497 taxa from the Baltic and the North Seas (Zettler et al., 2021). All well curated raw and aggregated data is currently stored in the open access repository together with the basic usage information. Here in the data descriptor we describe methods and algorithms used and provide details on metadata, structure and content of the dataset.

Our dataset can assist the studies where information on biomass has the central role by helping to more accurately translate WW into the more relevant AFDW. Data presented here are of use for a range of scientific studies, including:

(i) facilitating spatial and temporal comparison of secondary production and energy flow in marine ecosystems

(ii) assessment of species contribution to ecosystem functioning; supporting the generation of empirical models and predictive mapping of ecosystem services provided by marine benthic macroinvertebrates, by ensuring the most use of best taxonomic resolution and information on biomass

(iii) enabling user-defined sub-selection of data, that can be combined with own local data, instead of relying on single average number per large region

## 2 Materials and methods

Macrobenthic specimens were collected over the period from 1986 to 2020 in the Baltic and the North Sea (Fig. 1 and Table 1). Following HELOCM guidelines on sampling soft bottom macrofauna (HELCOM, 2017) most samples that were used for measurements included in the dataset were collected using Van Veen grab or 1-m dredge (type Kieler Kinderwagen). From hard-bottom habitats samples where partly derived by divers (Beisiegel et al., 2017). Routinely, samples were stored for at least three months before weighing. Biomass determination was carried out separately for each taxon. All nesting species like polychaetes or hermit crabs were removed from tubes or shells. *Molgula manhattensis*, a species of ascidian, and phoronids (represented solely by *Phoronis sp.*) require a special remark. As a rule, both taxa can hardly be separated from the glued grains of sand, which is why an exception has been made here. With these organisms the grains of sand were also commonly weighed in the laboratory routine. However, as desired, the AFDW only specifies the organic content, since sand and ash were deducted from that weight. Biomass of molluscs and echinoderms was measured with shells. The database only includes values based on individuals with wet weight exceeding 0.5 mg. The dry weight was estimated after drying the formalin material at 60°C to constant weight (for 12-24 hours, or longer, depending on material thickness). After



determination of dry weight, ash-free dry weight was measured following incineration at 500°C in a muffle furnace until weight constancy was reached. AFDW is recommended as the most accurate measure of biomass (Rumohr et al., 1987). Species nomenclature has been standardised in line with the World Register of Marine Species (WoRMS Editorial Board, 2021). In the continuous complementation of the database efforts were targeted to obtain sufficient number of measurements

for reliable estimates and cover as many frequent and characteristic species per region as possible (Table 2). The groups used in the dataset in order to facilitate the summary should be rather considered as functional, i.e. not strictly taxonomic, as they vary in rank ranging from Phylum to Order level. A word of caution should also be given regarding mean and confidence interval values reported in Table 2, calculated using R package 'DescTools' (Andri et mult. al., 2021) in R (R Core Team, 2013). Here we display the results based on all values of raw measurements of factors for all taxa included in the group.

Alternatively, depending on the aims and desired summary level, users are facilitated to obtain from the dataset mean values of conversion factors per group based on mean values per each taxon included in the group, thereby avoiding to overweight the reported statistics by dominant species, typically represented by high number of measurements.

### 3 Data availability and usage note

All measurements are available from IOW data repository: http://doi.io-warnemuende.de/10.12754/data-2021-0002 (Gogina

et al., 2021). We have included all quality-assured measurements values without prejudice. Reporting errors and updates of the data will be periodically issued. Users are encouraged to use the latest version of the data listed (under the 'versions' tab) at IOW repository. This contribution is based on data release 1.0. There are no limitations on the use of these data.

**Author contributions.** MG aided in data collection, adapted the dataset and prepared the paper with contributions from all co-

authors. AZ compiled and maintained the database and managed the quality assurance. MLZ secured funding, determined sampling strategies, conceived the investigation and ran the data collection campaigns.

**Competing interests.** The authors declare that they have no conflict of interest.

**Acknowledgements.** We gratefully acknowledge the work of all colleagues involved and responsible for field work and laboratory analysis, in particular Ines Glockzin, Frank Pohl, Stefanie Schubert, Tiffany Henschel, Sigrid Gründling-Pfaff and

Sarah Pirrung and all previous employee. We thank the captains and crews of RVs Elisabeth Mann Borgese, Poseidon, Alkor, Maria S. Merian and other vessels for their great support during multiple cruises. MG was partly funded by BMBF-Project MGF-Ostsee (03F0848A).

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

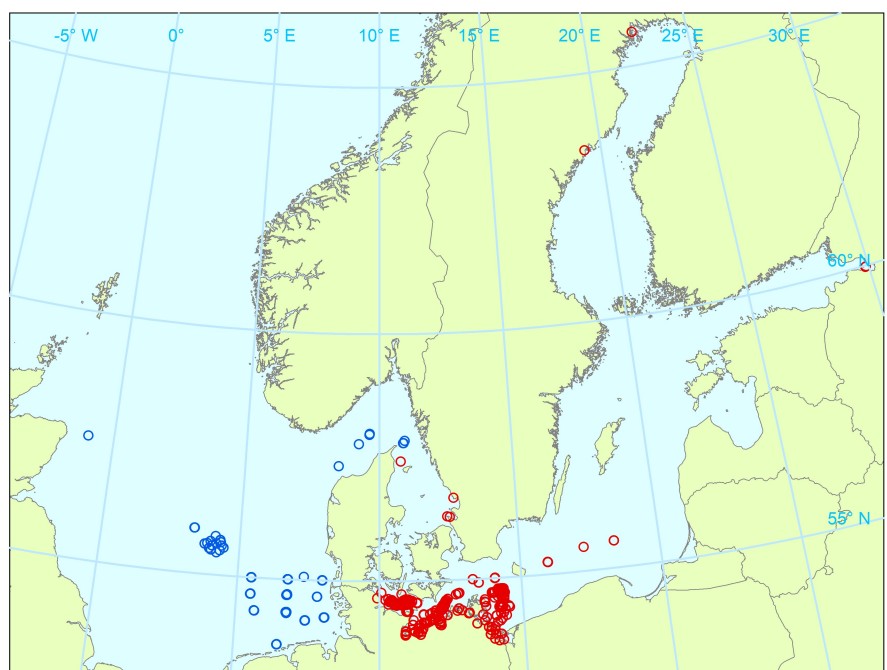

**Figure 1: The geographical locations of sites where individuals that reported measurements values are based on were collected.**
**Colour of symbols indicate habitats of the Baltic Sea (in red) and the North Sea (in blue). Data points may represent multiple observations at that locality. Projection: ETRS89 Lambert Azimuthal Equal-Area.**

**Table 1: Years of material collection and number of corresponding measurements per region included in the dataset.**

| Year | 1986 | 1987 | 1992 | 1993 | 1994 | 1995 | 1996 | 1997 | 1998 | 1999 | 2000 | 2001 | 2002 | 2003 | 2004 | 2005 | 2006 | 2007 | 2008 | 2009 | 2010 | 2011 | 2012 | 2013 | 2014 | 2015 | 2016 | 2017 | 2018 | 2019 | 2020 | Grand Total |
|---|---|---|---|---|---|---|---|---|---|---|---|---|---|---|---|---|---|---|---|---|---|---|---|---|---|---|---|---|---|---|---|---|
| Baltic Sea | | 1 | 243 | 517 | 662 | 428 | 13 | 234 | 134 | 279 | 219 | 328 | 193 | 301 | 387 | 247 | 417 | 398 | 336 | 385 | 392 | 492 | 1315 | 649 | 442 | 760 | 419 | 452 | 377 | 1351 | 439 | 12810 |
| North Sea | 1 | | | | | | | | | | | | | | | | | | 803 | 1103 | 1445 | 1657 | | | 29 | 7 | 8 | | | | | 5053 |





**Table 2: Weight-to-weight conversion factors for 29 major functional groups, differentiated by region, based on all raw values per taxa included in the group: AFDW = ash-free dry weight, WW = wet weight. DW = whole dry weight, CI = 95% confidence interval, N = number of values, SPP = number of species (taxa) per group.**

| Group | Baltic Sea | | | | North Sea | | | |
|---|---|---|---|---|---|---|---|---|
| | WW to DW (CI) | WW to AFDW (CI) | N | SPP | WW to DW (CI) | WW to AFDW (CI) | N | SPP |
| Amphipoda | 0.145 (0.142-0.149) | 0.143 (0.138-0.148) | 585 | 48 | 0.121 (0.118-0.124) | 0.128 (0.123-0.133) | 443 | 42 |
| Anthozoa | 0.187 (0.177-0.197) | 0.193 (0.181-0.206) | 103 | 8 | 0.13 (0.123-0.137) | 0.141 (0.128-0.155) | 77 | 4 |
| Arachnida | 0.242 (0.218-0.267) | | 20 | 1 | 0.215 (0.19-0.239) | | | |
| Ascidiacea | 0.178 (0.14-0.215) | 0.15 (-0.146-0.446) | 108 | 4 | 0.045 (0.04-0.05) | 0.012 (0.006-0.018) | 4 | 3 |
| Bivalvia | 0.489 (0.484-0.494) | 0.473 (0.465-0.482) | 4391 | 42 | 0.073 (0.072-0.074) | 0.084 (0.081-0.087) | 1063 | 39 |
| Bryozoa | 0.161 (0.146-0.176) | | 30 | 2 | 0.073 (0.064-0.082) | | | |
| Caudofoveata | | 0.269 (0.246-0.293) | | | | 0.189 (0.133-0.245) | 11 | 1 |
| Cirripedia | 0.495 (0.474-0.516) | 0.649 (0.575-0.723) | 60 | 4 | 0.052 (0.046-0.058) | 0.083 (0-0.171) | 5 | 3 |
| Cumacea | 0.156 (0.153-0.158) | 0.152 (0.134-0.169) | 541 | 3 | 0.12 (0.117-0.122) | 0.13 (0.112-0.147) | 54 | 9 |
| Decapoda | 0.192 (0.182-0.201) | 0.181 (0.167-0.195) | 106 | 10 | 0.142 (0.137-0.147) | 0.119 (0.113-0.126) | 127 | 20 |
| Echinodermata | 0.35 (0.33-0.37) | 0.404 (0.392-0.417) | 197 | 6 | 0.071 (0.067-0.076) | 0.077 (0.071-0.082) | 382 | 13 |
| Gastropoda | 0.463 (0.452-0.473) | 0.617 (0.601-0.632) | 787 | 55 | 0.106 (0.103-0.11) | 0.096 (0.089-0.102) | 260 | 14 |
| Hirudinea | 0.193 (0.103-0.284) | | 6 | 5 | 0.178 (0.089-0.267) | | | |
| Hydrozoa | 0.164 (0-0.512) | | 2 | 2 | 0.099 (0-0.235) | | | |
| Insecta | 0.149 (0.127-0.171) | | 31 | 5 | 0.12 (0.098-0.141) | | | |
| Isopoda | 0.176 (0.167-0.185) | 0.235 (0.164-0.307) | 154 | 12 | 0.119 (0.112-0.125) | 0.221 (0.149-0.294) | 7 | 3 |
| Leptocardii | | 0.143 (0.13-0.157) | | | | 0.134 (0.121-0.147) | 12 | 1 |
| Mysida | 0.15 (0.145-0.155) | 0.167 (0.154-0.18) | 128 | 8 | 0.131 (0.125-0.138) | 0.154 (0.141-0.168) | 29 | 2 |
| Nemertea | 0.159 (0.154-0.164) | 0.174 (0.166-0.182) | 282 | 6 | 0.142 (0.138-0.147) | 0.158 (0.15-0.166) | 199 | 5 |
| Oligochaeta | 0.154 (0.148-0.159) | 0.28 | 363 | 11 | 0.129 (0.125-0.134) | 0.256 | 1 | 1 |
| Phoronida | 0.74 (0.723-0.757) | 0.544 (0.513-0.574) | 33 | 1 | 0.027 (0.016-0.038) | 0.069 (0.061-0.077) | 69 | 1 |
| Platyhelminthes | 0.165 (0.151-0.178) | 0.105 (0.08-0.131) | 27 | 1 | 0.144 (0.131-0.157) | 0.095 (0.07-0.121) | 11 | 1 |
| Polychaeta | 0.168 (0.166-0.17) | 0.189 (0.185-0.192) | 4490 | 93 | 0.119 (0.117-0.12) | 0.148 (0.145-0.15) | 2293 | 93 |
| Polyplacophora | 0.465 (0.434-0.497) | | 6 | 1 | 0.105 (0.09-0.12) | | | |
| Porifera | 0.109 (0.097-0.122) | | 51 | 3 | 0.057 (0.049-0.065) | | | |
| Priapulida | 0.118 (0.115-0.122) | | 269 | 2 | 0.106 (0.103-0.109) | | | |
| Pycnogonida | 0.142 (0.127-0.157) | 0.186 (0.112-0.261) | 22 | 2 | 0.107 (0.092-0.121) | 0.166 (0.097-0.235) | 3 | 1 |
| Sipuncula | | 0.166 (0.091-0.24) | | | | 0.148 (0.057-0.238) | 3 | 1 |
| Tanaidacea | 0.196 (0.16-0.231) | | 18 | 4 | 0.151 (0.12-0.183) | | | |
| Overall | | | 12810 | 339 | | | 5053 | 257 |