# Peer review of "Weight-to-weight conversion factors for benthic macrofauna: recent measurements from the Baltic and the North Seas"

_Earth System Science Data, 2021_

## Author Comment (AC2)

**Response to Referees' comments**

We would like to greatly thank Mats Blomqvist for the comprehensive review and helpful and inspiring comments. We think the manuscript has improved considerably. On the base of the remarks we have edited and corrected the dataset, text file
5 with supplementary and data description text.

Referees' comments:

This is a really useful and important publication. The data published represents a significant amount of work now accessible for all to reuse.

10 There is a strong dominance of data from SW Baltic waters and to better represent Baltic Sea and North Sea it is my hope that the dataset will be enlarged in the future, possibly in cooperation with other laboratories.

In the supplementary information wordfile with descriptions of the data set column headings there is an error for column Group. Description needs to be changed as it is a copy of the previous column description.

15 Response to referee's comment No.1: *We have corrected the column description accordingly. It now states: "group the taxa is classified to in order to facilitate summary (these groups are rather functional, i.e. not strictly taxonomic, as they vary in rank ranging from Phylum to Order level)". For consistency all over the text we have also corrected "wet weight", "ash free" and "dry weight" to "wet-weight", "ash-free" und "dry -weight".*

20 In the actual excel data file there are two records classified as Baltic that are situated in the middle of the North Sea (station OB20 and OB40).

Response to referee's comment No.2: *Indeed, both stations are in the North Sea (Doggerbank). We have corrected the dataset accordingly, and assigned the corresponding factors to the correct Region. We have also edited Tables 1 and 2 by correcting the corresponding values. Moreover, in Table 2 we found and corrected another error – values for Baltic Sea WW*
25 *to AFDW (CI) factors and North Sea WW to DW (CI) factors were swapped. In the dataset itself we have also corrected the remark on the "Raw_Single_Measurments_Factors" Sheet (Cell A2) to more accurate formulation: "Values only from measurements (individuals or pooled individuals) with wet-weight >= 0.5 mg. The DOI stated in the manuscript was updated (to http://doi.io-warnemuende.de/10.12754/data-2021-0002-01) and points now to the updated version of the dataset (with the remark that it is New Version Of 10.12754/data-2021-0002).*

30

The manuscript is well written and contains a good and clear description of the dataset. There is a small need to improve the English in order to make some sentences easier to read. Here are line numbers were improvements are needed (in most cases also with suggestions for change):

7-9 this sentence is hard to understand, reformulate

Response to referee's comment No.3: *We have corrected the text accordingly: "Availability of standardized biomass data is essential for studying population dynamics, energy flows, fisheries and food web interactions. To make the estimates of biomass consistent weight-to-weight conversion factors are often used, for example to translate more widely available measurements of wet-weights into required dry-weights and ash-free dry-weights metrics."*

12 The dataset provides a broad

Response to referee's comment No.4: *We have corrected the text accordingly.*

15 The dataset can thereby be used to quantify... Delete ", and refine current ecological theory" since this is a bit too much

Response to referee's comment No.5: *We have corrected the text accordingly.*

18 , and the role of

Response to referee's comment No.6: *We have corrected the text accordingly.*

22-23 Conversion factors... this sentence is not understandable, seems to be incomplete?

Response to referee's comment No.7: *We have corrected the sentence to: "Conversion factors are derived from subsamples to enable data standardisation and determination of dry-weight for large volume of material."*

24 pioneered in publication of compilations

Response to referee's comment No.8: *We have corrected the text accordingly.*

27 though allowing... Move comma from after estimates to after groups

Response to referee's comment No.9: *We have corrected the text accordingly.*

30 includes only a little share

Response to referee's comment No.10: *We have corrected the text accordingly.*

32 the importance of the present compilation

Response to referee's comment No.11: *We have corrected the text accordingly.*

65 36 repository together with basic usage .... Here in the data description we

Response to referee's comment No.12: *We have corrected the text accordingly.*

38 biomass has a central role

Response to referee's comment No.13: *We have corrected the text accordingly.*

70

44-45 this point does not fit in the list of scientific study types. Text can be merged into row 23-24 instead.

Response to referee's comment No.14: *We have corrected the text accordingly.*

62 In the continous .... This sentence can be simplified. No suggestion on how.

75 Response to referee's comment No.15: *We have tried to simplify the sentence, that is now written as: "The database is continuously enlarged, with main efforts targeted to obtain sufficient number of measurements for reliable estimates and to cover as many characteristic species per region as possible (Table 2)."*

74 the data will be done periodically.

80 Response to referee's comment No.16: *We have corrected the text accordingly.*

74 there is no version tab in the web page (link at row 74)

Response to referee's comment No.17: *We have removed the remark "(under the 'versions' tab)" and reformulated the sentence: "
[revised manuscript text omitted]

140

| | | | | | | | | |
|---|---|---|---|---|---|---|---|---|
| Priapulida | 0.118 (0.115-0.122) | 0.106 (0.103-0.109) | 269 | 2 |  | | | |
| Pycnogonida | 0.142 (0.127-0.157) | 0.107 (0.092-0.121) | 22 | 2 | 0.186 (0.112-0.261) | 0.166 (0.097-0.235) | 3 | 1 |
| Sipuncula | |  | | | 0.166 (0.091-0.24) | 0.148 (0.057-0.238) | 3 | 1 |
| Tanaidacea | 0.196 (0.16-0.231) | 0.151 (0.12-0.183) | 18 | 4 |  | | | |
| Overall | | |  12808 |  337 | | |  055 |  259 |